# Proteomic Alteration in the Progression of Multiple Myeloma: A Comprehensive Review

**DOI:** 10.3390/diagnostics13142328

**Published:** 2023-07-10

**Authors:** Nor Hayati Ismail, Ali Mussa, Mutaz Jamal Al-Khreisat, Shafini Mohamed Yusoff, Azlan Husin, Muhammad Farid Johan

**Affiliations:** 1Department of Haematology, School of Medical Sciences, Universiti Sains Malaysia, Kubang Kerian 16150, Kelantan, Malaysia; hayatiismail255@gmail.com (N.H.I.); alimaha1341989@gmail.com (A.M.); mutaz.alkhreisat@student.usm.my (M.J.A.-K.); shafini@usm.my (S.M.Y.); 2Department of Biology, Faculty of Education, Omdurman Islamic University, Omdurman P.O. Box 382, Sudan; 3Department of Internal Medicine, School of Medical Sciences, Universiti Sains Malaysia, Kubang Kerian 16150, Kelantan, Malaysia; azlanh@usm.my

**Keywords:** multiple myeloma, proteomics, post-translational modifications, mass spectrometry

## Abstract

Multiple myeloma (MM) is an incurable hematologic malignancy. Most MM patients are diagnosed at a late stage because the early symptoms of the disease can be uncertain and nonspecific, often resembling other, more common conditions. Additionally, MM patients are commonly associated with rapid relapse and an inevitable refractory phase. MM is characterized by the abnormal proliferation of monoclonal plasma cells in the bone marrow. During the progression of MM, massive genomic alterations occur that target multiple signaling pathways and are accompanied by a multistep process involving differentiation, proliferation, and invasion. Moreover, the transformation of healthy plasma cell biology into genetically heterogeneous MM clones is driven by a variety of post-translational protein modifications (PTMs), which has complicated the discovery of effective treatments. PTMs have been identified as the most promising candidates for biomarker detection, and further research has been recommended to develop promising surrogate markers. Proteomics research has begun in MM, and a comprehensive literature review is available. However, proteomics applications in MM have yet to make significant progress. Exploration of proteomic alterations in MM is worthwhile to improve understanding of the pathophysiology of MM and to search for new treatment targets. Proteomics studies using mass spectrometry (MS) in conjunction with robust bioinformatics tools are an excellent way to learn more about protein changes and modifications during disease progression MM. This article addresses in depth the proteomic changes associated with MM disease transformation.

## 1. Introduction

Multiple myeloma (MM) is defined as the uncontrolled and aberrant proliferation of monoclonal plasma cells in the bone marrow, resulting in hypercalcemia, anemia, renal insufficiency, and bone lesions [1]. MM accounts for 1% of neoplastic diseases and >10% of all hematologic malignancies, with the elderly population being the most affected [2,3]. Myeloma is a genetically complex and highly heterogeneous disease that progresses through a succession of phases to malignancy, with genetic accumulation being reversed over time [4]. Various genetic aberrations found in MM cells, such as chromosomal translocations, aneuploidy [5,6], somatic mutations [7,8], and epigenetic abnormalities [9], are fundamental events involved in MM pathogenesis. MM evolves from asymptomatic and precancerous precursors, monoclonal gammopathy of undetermined significance (MGUS) and smoldering MM (SMM), to symptomatic MM, including intramedullary and extramedullary myeloma [10]. Compared to the past 15 years, MM survivability has improved significantly with a median survival rate of 55.6% after five years [11]. The development of drugs with novel mechanisms of action [12] such as immunomodulatory agents (IMiDs), proteasome inhibitors (Pis), and monoclonal antibodies [13], is responsible. However, MM is still considered an incurable disease that poses enormous challenges to treatment. Most patients with MM invariably suffer from a rapid relapse and refractory conditions [14], despite the tremendous discovery and approval of innovative drugs for therapy [15,16].

The integration of MM with the increased complexity of the genome poses significant challenges and burdens for the management of patients. Therefore, extensive research to discover new and specific biomarkers to diagnose and assess patient prognosis in MM is urgently needed. In this review, we focus on the application of proteomics technology, which offers significant breakthroughs in the study of protein identification and potential biomarkers that may be involved in MM progression.

## 2. Proteomics Breakthroughs in Protein Identification

In recent years, the application of multi-omics analysis in translational research on patient samples has proven to be a robust technique. Multiple -omics approaches, including genomics, proteomics, metabolomics, and transcriptomics, have contributed to extensive insights in integrated study of patient data [17]. Proteomics is the integration of powerful bioanalytical protein studies [18] on specific types of cells (e.g., tumour, blood, or tissue) in a wide range of diseases. Alteration of protein expression levels and/or post-translational modification allows for the investigation of potential protein biomarkers for diagnostic purposes and new therapeutic discoveries [19,20]. To date, the outcome of a combination of proteomics and genomics assays has shown promise in translational disease research and could lead to future advances in next-generation diagnostic and therapeutic techniques. The invention of a robust protein separation technology, two-dimensional gel electrophoresis (2D-PAGE) [21], and advancements in mass spectrometry techniques have enabled accurate mass and chemical structure analysis-propelled proteomics research [22]. The proteomics workflow includes protein extraction from the biological sample, quantification of protein content via Bradford assay, protein separation via first-dimension electrophoresis (isoelectric focusing) based on its isoelectric points, and second-dimension electrophoresis, which separates proteins based on their molecular weight. The gel is then stained (e.g., silver staining). Protein spots with significant differences in expression are proteolytically cleaved using the trypsin enzyme, followed by mass spectrometry and software analysis to identify the target proteins. Subsequently, the rapid expansion of shotgun proteomics has contributed to the hypersensitivity of mass spectrometry by supplanting the laborious components of 2D gel-based proteomics [23]. Moreover, the fluorescence-based two-dimensional difference gel electrophoresis (2D-DIGE) technique represents a significant advancement over the conventional 2D-PAGE method, as it enables advanced quantitative analysis of proteomic samples. The 2D-DIGE technique is characterized by its robust sensitivity, improved accuracy, and resolution, which are achieved through the pre-labelling of individual protein samples with fluorescent cyanine dyes (Cy2, Cy3, and Cy5). Following this, the labelled samples are mixed together and subjected to 2-dimensional electrophoresis (2DE), after which the gel is visualized using fluorescence imaging to identify variations in protein expression across the samples [24], as illustrated in Figure 1.

Due to the widespread application of large-scale genome research, several databases for an enormous number of peptide and protein sequences are currently under development [25]. In parallel, numerous bioinformatics tools have been developed to alleviate the huge output of MS data for the identification of proteins and their biological or pathogenic PTMs [26]. The emergence of functional proteomics is a result of breakthroughs in protein identification enabled by quantitative mass spectrometry, biofluid array technology, phosphoproteomics, single-cell proteomics, and glycoproteomics. Several proteomic technologies for protein profiling are summarized in Table 1. These ideal approaches focused on the characterization of enzymatic activities, protein–protein interactions and networking, and post-translational modification of proteins and protein clusters at the proteome level [27]. Proteomics applications can bridge the gap between cytogenetics, epigenetics, and functional gene expression, all of which focus on genetic abnormalities. Remarkably, significant changes in gene expression are not necessarily reflected at the protein level. This is because prior to the production of functional proteins, mRNA needs several regulatory processes such as RNA metabolism, ribosomal protein synthesis, post-translational modifications, and protein clearance. In addition to genomic regulation, multiple fully folded proteins fuel critical functional biological pathways [28].

The utilization of mass spectrometry-based proteomics has emerged as a potent mechanism for the precise evaluation of proteins and their alterations in various diseases. The technique of label-free quantification (LFQ) is a frequently employed method in proteomics based on mass spectrometry. It can be executed through two distinct modes of acquisition, namely, data-independent acquisition (DIA) and data-dependent acquisition (DDA) [29]. The LFQ-DIA methodology is a quantitative proteomics technique that entails the methodical fragmentation of precursor ions originating from all peptides within a specified mass-to-charge (*m*/*z*) range. The DIA methodology involves partitioning the complete mass spectrum into discrete intervals, followed by sequential fragmentation of each interval. The LFQ-DIA methodology enables the thorough and consistent quantification of peptides and proteins in a complex sample by obtaining fragment ion spectra across a spectrum of *m*/*z* values. Quantification is commonly accomplished through the extraction of ion intensities from particular fragment ions that serve as representations of the peptides under scrutiny. LFQ-DIA has been observed to provide several benefits over alternative quantification techniques, including but not limited to enhanced reproducibility, greater peptide coverage, and decreased rates of missing values. This approach proves to be highly advantageous in the examination of extensive sample cohorts and intricate mixtures [30]. The LFQ-DIA technique has been efficaciously employed in various studies with the aim of detecting proteins that are differentially expressed, delineating biomarkers that are specific to the disease, and investigating the molecular mechanisms that underlie the pathogenesis of the disease [31]. The LFQ-DDA method is a commonly employed technique for label-free quantification in proteomics using mass spectrometry. The process of selecting precursor ions for fragmentation in DDA is based on their intensity or abundance. In general, the precursor ions exhibiting the highest intensity in a survey scan are selectively subjected to fragmentation, leading to the production of MS/MS spectra. Subsequently, these spectra are employed for the purpose of identifying and quantifying peptides [30]. LFQ-DDA presents certain benefits, including heightened sensitivity in the identification of peptides and the ability to be utilized in conjunction with a high-resolution mass spectrometry apparatus. Nevertheless, it exhibits restricted reproducibility and dynamic range in comparison to DIA techniques [30]. The employment of both LFQ-DIA and LFQ-DDA methodologies is deemed advantageous for label-free quantification in the context of mass spectrometry-based proteomics analysis. LFQ-DIA yields a more extensive and consistent quantification outcome, whereas LFQ-DDA exhibits heightened sensitivity in terms of peptide identification. The selection of methodology is contingent upon the particular research inquiry, the intricacy of the sample, and the accessibility of instrumentation [32].

The Cell Surface Capturing (CSC) technique is a significant proteomic approach that focuses on the targeted enrichment and analysis of proteins that are specifically located on the surface of cells [33]. Proteomes located on the surface of cells are of significant importance in facilitating cellular signaling, communication, and engagement with the extracellular milieu. The utilization of cell surface proteins in CSC enables the discernment and delineation of pivotal agents implicated in diverse cellular mechanisms and pathological routes [34]. The identification of therapeutic targets on the cell surface is a critical aspect in the development of efficacious treatments for multiple myeloma. CSC has been used to identify a number of cell-surface proteins that exhibit high expression levels in multiple myeloma cells and may serve as promising therapeutic targets. These proteins include CD38, B-cell maturation antigen (BCMA), SLAMF7, CD138 [35,36,37], CD56, CD46, CD74, CD48, LY9, and integrins [38]. Patiño-Escobar et al. recently made modifications to the glycoprotein CSC technique by utilizing cell-impermeant reagents to oxidize and biotinylate N-linked glycoproteins present on the surface of live cells. The process of enriching surface proteins involves the use of streptavidin beads, followed by on-bead proteolysis to convert the samples into peptides that can be analyzed using mass spectrometry. They documented the proteomic profile of the surface of MM cells, identifying novel targets for immunotherapy and markers of resistance to therapeutic agents [39].

The field of biomedical research has recently witnessed the emergence of a state-of-the-art methodology known as single-cell proteomics [40]. The utilization of this method represents a potent approach to deconstructing the cellular and molecular terrain at the level of individual cells, as opposed to furnishing aggregated outcomes. The utilization of single-cell technologies holds the potential to resolve unresolved inquiries in the domain of myeloma biology and has significantly transformed our comprehension of inter- and intra-tumor heterogeneity, the tumor microenvironment, and the underlying mechanisms of therapeutic resistance in MM [41]. Additionally, the utilization of Machine Learning (ML) and Deep Learning (DL) algorithms presents a potent approach to scrutinizing intricate data emanating from single-cell proteomics experiments. ML algorithms are designed to learn patterns and relationships from input data, enabling them to make predictions or classifications [42]. Furthermore, the integration of DL incorporates the capability to extract high-level features from raw data over multiple stages. DL is a suitable method for analyzing single-cell omics data, which is typically characterized by high dimensionality, sparsity, and complexity [43].

## 3. Post-Translational Modifications in MM

Proteins interact with one another to execute a variety of biological mechanisms [44]. As a result of the dynamic interaction between cellular homeostasis mechanisms and protein interactions that enable speed, adaptability, and compartmentalization, specific proteins are synthesized in the meticulous subcellular compartment and activated in response to stimulation by specific signals [45]. Proteins undergo PTMs, which involve phosphorylation, with a focus on serine, tyrosine, and threonine residues; glycosylation; and ubiquitination. Titanium dioxide enrichment of phosphopeptides and LC-MS/MS analysis revealed 530 phosphorylation sites in primary MM cells from patients. These phosphorylation sites are spread across 325 different phosphopeptides and 260 proteins [46]. Ge et al. studied the cytotoxic effect in MM after bortezomib therapy using quantitative serine/threonine phosphoproteomics. A total of 233 phosphorylated proteins were found, 72 of which were associated with significant upregulation of 1.5-fold or more after bortezomib treatment. Bortezomib induces Ser38 phosphorylation on stathmin, resulting in unstable microtubules and the eventual death of MM cells [47]. Another study found that a fibroblast growth factor receptor 3 (FGFR3) inhibitor reduced the phosphorylation of 52 proteins with 61 pY-phosphotyrosine (pY) sites in KMS11-MM cell lines. Since FGFR3 overexpression in patients with t(4;14) translocation is associated with chemoresistance and a poor prognosis, they might be advantageous as downstream FGFR3 targets [48]. M-proteins have a greater rate of glycosylation, notably in the fragment antigen-binding (Fab) and fragment crystallizable (Fc) domains [49]. Glycosylation modulates several events in the Fab region, including antigen attachment, immune complex formation, and immunoglobulin half-life extension [50]. In contrast, N-glycosylation of the Fc region occurs at asparagine 297, strengthens MM with a stable structure to improve receptor binding on effector cells, and regulates immunoglobulin activity overall [51]. As a result of these outstanding outcomes, the glycoproteomic profile of immunoglobulin has the potential to be used as a possible biomarker for MM.

Ubiquitination maintains the functional state of proteins in homeostasis through an enzymatic cascade comprising ubiquitin-activating enzymes (E1), ubiquitin-conjugating enzymes (E2), and ubiquitin-protein ligases (E3) [52]. Due to the high amount of monoclonal antibodies synthesized in MM, the delicate balance between protein synthesis, folding, and degradation is essential for the survival of MM cells [53]. Covalent interactions between E3 ligase and ubiquitin promote monoubiquitination or polyubiquitination. K48 linked polyubiquitination is the most well-studied polyubiquitination pathway related to proteasomal degradation via the ATP-dependent 26 S-proteasome complex [52]. K-63 is associated with endocytotoxic migration, inflammation, and DNA damage repair. Another study has found that K11-linked proteins are involved in mitotic regulation and endoplasmic reticulum-associated degradation.

Monoubiquitination enhances protein activity and translocation errors [54]. Monoubiquitination occurring at histones H2A and H2B derived from E3-ligases of the polycomb repression complex (BMI1 (B cell-specific Moloney murine leukemia virus integration site 1), RING1A/RING1B (Really Interesting New Gene 1A/1B), PCGF2 (Polycomb group RING finger protein 2), RNF8, and 2A-HUB/hRUL138 inhibited RNA polymerase 2 elongation activity and eventually suppressed gene expression [55,56,57,58]. Concurrently, higher BMI1 expression has been observed in the plasma cells of patients with MM [59,60], as well as increased MM cell proliferation in vitro and in vivo [61]. In MM, aberrant ubiquitination is predominantly described in the NFB pathway [62], which leads to more aggressive MM by obstructing the deubiquitinating enzyme CYLD [63]. Activation of the NFB pathway regulates the apoptotic mechanism and produces a genetic deletion in MM, such as homozygous deletions of TRAF3, which is induced by the ubiquitin ligases BIRC2/BIRC3 [64,65].

An extensive effort has been made to formulate certain antibodies for the purpose of post-translational modification analysis. Nevertheless, this effort is challenged by the low binding affinity of available antibodies due to the small size of post-translational modification (PTM) motifs. Additionally, chemical structure similarities among certain PTMs, inadequate antigenicity, and other obstacles in antibody production [66] are also credited with the limitation. Antibodies that are specific to pan-PTM are frequently utilized for immunoaffinity enrichment before conducting LC-MS/MS, Western blotting, protein microarrays, immunohistochemistry, and flow cytometry. The aforementioned methodology has been efficaciously employed for the comprehensive evaluation of protein lysine acetylation [67,68], arginine methylation [69], tyrosine nitration [70], and tyrosine phosphorylation [71,72]. Additionally, peptide PTM motif-specific antibodies have been utilized to recognize kinase substrates and measure their PTM alterations [73]. The approach for detecting new substrates can be extended to cases where a PTM regulatory enzyme of interest possesses a consensus PTM motif. However, several other post-translational modifications (PTMs) encounter challenges in generating site-specific antibodies, either because of the PTM’s size or swiftness (such as ubiquitination), or due to inadequate evidence of site occupancy, which discourages investing in antibody generation [74].

In numerous proteomics investigations, post-translational modifications (PTMs) are labelled through chemical derivatization, which involves the incorporation of affinity tags to enhance the abundance of PTM peptides that are of interest. For example, azide was utilized as a chemical label for post-translational modification (PTM) proteins and then linked to an affinity linker, such as biotin [75,76]. This technique was employed in the detection of protein farnesylation [77], O-GlcNAc modifications [78], palmitoylation [79], and myristoylation [80].

Although various components of the proteomics approach have progressed in the meantime, the process of protein digestion remains predominantly reliant on a single enzyme, typically trypsin. Other proteases, including chymotrypsin, LysC, LysN, AspN, GluC, and ArgC, are utilized to a certain extent [81]. Due to their unique specificities, these proteases produce distinct sets of peptides. Consequently, subjecting the same proteome to parallel digestion with multiple proteases can cover complementary regions of the protein sequence space and the proteome itself [82,83]. The protease-digested lysate is an ideal strategy to increase the sequence coverage, thereby facilitating the differentiation of closely associated protein isoforms. Additionally, it enables accurate and reliable protein identification and quantification, encompassing diverse post-translational modification (PTM) sites [84,85].

## 4. Proteomics Alteration in MM

### 4.1. Monoclonal Gammopathy of Undetermined Significance (MGUS)

MGUS is an early precursor of clonal plasma cell proliferation that forecasts the development of MM [86]. MGUS is an asymptomatic, incidentally diagnosed disorder that is evaluated for progression to MM using a serum protein electrophoresis profile or immunofixation (SPEP or IFX). MGUS is diagnosed when there are less than 30 g/L of monoclonal protein (M-protein) in the blood or less than 10% plasma cells in the bone marrow [87]. In MGUS, 11 proteins were identified as core matrisome proteins, including bone marrow proteoglycan 2 (PRG2) and bone marrow proteoglycan 3 (PRG3), and the remaining 9 proteins were associated with the matrisome, including the ECM-affiliated protein ficolin 1 (FCN1), the ECM-remodeling enzymes CTSG and Serpins, and the secreted factors HRNR, S100A8, and S100. In contrast, 32 proteins were discovered in MM patients, including 10 core matrisome proteins and 22 matrisome-associated proteins. Curiously, Annexin A2 (ANXA2) and Galectin-1 (LGALS1) are potential biomarkers in MM, since both were not observed in bone marrow from control donors or MGUS patients [88]. ANXA2 is also involved in cell development as well as promoting osteoclast synthesis and bone erosion, resulting in an immunosuppressive microenvironment in MM [89]. It also helps MM cells grow and escape from the apoptotic pathway in MM cell lines [90].

The proteomic analysis of proteins obtained from circulating exosomes in 5 MGUS and 10 MM patients and 45 healthy subjects was investigated by Manier et al. using mass spectrometry. They identified a total of 272 proteins in the circulating exosomes. These proteins include annexins, CD9, HSP70, and Rab proteins (Rab7a, Rab5, and Rab27b), which are among those strongly linked with exosomes. The peptide counts for fibronectin, AMBP protein, and Ig gamma-1 chain C region were found to be markedly different. Interestingly, they also reported that elevated expression of fibronection in the microenvironment of multiple myeloma has been associated with both tumor progression and resistance to drug therapies [91].

Proteomic evaluation of all MGUS, SMM, and MM patients revealed a linear trend in vascular endothelial growth factor A (VEGF-A), vascular endothelial growth factor 2 (VEGFR2), and interleukin-6 (IL-6) levels across MM stages. VEGFR2 downregulation was observed with the courses of MGUS, SMM, and MM [92]. Furthermore, the VEGFR2-604TT genotype is more common and associated with MM patients with a higher disease burden. Vascular endothelial growth factor (VEGF) was found to be more prominent in both MM cell lines and primary MM cells from the bone marrow during cancer angiogenesis [93].

### 4.2. Smoldering Multiple Myeloma (SMM)

MGUS precursors underwent extensive mutational events at a magnitude of 0.1 to 10 per megabase, along with genetic alterations that affected one or more genes, including aneuploidy, chromosomal translocations, single nucleotide variants, small insertions and deletions, and copy number variants in the state transition to SMM [94,95,96]. It was reported that the higher progression risk of SMM to MM was 10% per year, whereas the progression risk of MGUS was only 1% [97].

Fernandez et al. explored the potential interaction between immunologic and pathologic patterns in SMM by performing proteomic profiling with mass cytometry, T cell receptor (TCR) sequencing, and serologic assays. As a result, the similar traits exhibited in these three assays were further divided into three taxa [98]. Proteomic analysis discovered cytokines and other soluble compounds of interest associated with immune cells in SMM patients. Taxon 1 is characterized by the same traits as MGUS and SMM, with elevated levels of pleiotrophin and NK cells and decreased levels of CD8+ T cells in the innate or transitional immunological milieu [98]. Similarly, Chen et al. discovered an increased pleiotrophin level in MM cells, which was linked to myeloma progression [99]. Furthermore, Yeh et al. discovered that pleiotrophin was higher in MGUS serum samples as determined by an indirect enzyme-linked immunosorbent assay (ELISA) test [100]. In contrast, both Taxon 2 and 3 exhibited reduced pleiotrophin levels, suggesting a lower disease burden. Taxon 2 was related to increased CD57+CD4+ T cells, which indicated an active T cell immunological state, as well as Granzyme H, which distinguished SMM from healthy bone marrow. Taxon 3 is also related to greater levels of central memory/naive CD4+ and CD8+ T cells, as well as apoptosis-associated steering components such as Gal-1, HO-1, TRAIL, and TWEAK, and immune suppression components such as Arg-1 and Gal-9 [98]. Interestingly, the discovery of distinct tumor microenvironment (TME) heterogeneity distinguishes not just MM and its clinical precursors, but also SMM itself. Integrative approaches increased discriminatory power in revealing the intricate cellular and molecular profile of the MM TME in early disease stages, potentially leading to improved risk assessment and therapy. In addition, another study reported that a comprehensive proteomics analysis identified 1302 dysregulated proteins, the majority of which are increased in tandem with MM advancement. Further protein identification has revealed that both the MGUS and SMM stages are associated with an increase in the expression of proteins that govern proteotoxic stress and osteolysis [101].

### 4.3. Multiple Myeloma (MM)

Integrative approaches using matrix-assisted laser desorption/ionization time-of-flight mass spectrometry coupled with the ClinProt system revealed ten significantly different expressed proteins. Six and four of them were upregulated and downregulated, respectively, in MM patients compared to healthy controls. Four downregulated proteins were identified: alpha-fetoprotein (AFP), component 3f (C3f), fibronectin 1 (FN1), and glutathione S-transferase π 1 (GSTP1) [102]. AFP is a glycoprotein and member of the serum albumin gene family, elevated levels of which are frequently observed in patients with 47 various malignancies and non-cancerous disorders [103]. Yang et al. published a case study demonstrating an increased AFP level in a 66-year-old male patient diagnosed with IgD-light chain (Stage III B) where the patient also revealed extensive bone destruction of the skull [103].

C3f is an essential component for activating the human innate immune system. It is also responsible for the elimination of potential pathogens [104] and plays a crucial role in the complement cascade [105]. The presence of an elevated C3f level is linked to an increased risk of developing renal illnesses, metabolic syndrome, and rheumatological diseases [106,107]. Due to the limited number of publications, additional study is required in the case of MM to define the association between C3f and MM pathogenesis. Upregulation of FN1 is associated with a poor prognosis in certain cancers, including thyroid cancer and nasopharyngeal cancer [108,109], because of greater cell motility, cytoskeletal structure, and oncogenic transformation, as previously described in multiple types of cell lines [110,111]. Following FN1 overexpression, apoptotic mechanisms via the NF-B pathway are inhibited, which promotes tumor cell migration and invasion [112]. FN1 is recommended as a strong biomarker for drug resistance and radiotherapy in head and neck squamous cell cancer [113]. As a result of its identically elevated profile in MM, FN1 could be employed as a possible biomarker in diagnosing MM [114]. Glutathione S-transferases (GSTs) are phase-II metabolic enzymes [115] that regulate the antioxidant response system [116], tumorigenesis, and detoxification [117] and have been found to be increased in a variety of cancers, including drug-resistant cell lines [118]. GSTs regulate many kinase pathways to promote tumorigenesis [115,119]. In MM, GST regulates the BM environment by causing aberrant redox, which results in the initiation of myeloproliferative processes [120]. GST upregulation may serve as a possible prognostic indication in MM and as a robust indicator for assessing patients’ MRD [102].

Proteome profiling of MM mononuclear cells CD138-CD105 on newly diagnosed MM (NDMM) patients revealed 66 elevated proteins, the majority of which are involved in energy metabolism, cellular compartmentalization, and protein production. In contrast, nine downregulated protein locations are associated with immune response, particularly innate immunity, which supports the hypothesis of an immune response evasion route in MM. Conversely, 12 upregulated and two downregulated proteins were discovered in MM-mesenchymal stem cells, with the most important upregulated proteins being S100-A11, calnexin, proteasome subunit alpha type-1, and RAB7A protein. Interestingly, each of the identified proteins has been linked to various factors involved in cancer pathogenesis [121]. Dytfeld et al. performed proteome profiling on plasma cells from naïve MM patients and also discovered an alternation in energy metabolism pathways that was associated with a satisfactory response to bortezomib-based therapy regimens [122].

The extracellular matrix of the bone marrow in patients with MM has been intriguingly characterized using proteomic profiling. ANXA2 and LGALS1 were shown to be elevated in NDMM patients, and the same trend was observed in MM cell lines [90,123]. These proteins are thought to be crucial for regulating survival in MM. ANXA2 has been long reported to be significantly upregulated in multiple myeloma in comparison to normal plasma cells [124]. Moreover, the ANXA2 receptor, also known as AX2R [125], exhibits augmented expression in primary myeloma cells as well as in cell lines, thereby promoting the growth of myeloma cells and their adherence to stromal cells [126]. ANXA2 has been found to facilitate multiple pathways, such as angiogenesis [127], osteoblastic mineralization [128], and the growth and differentiation of osteoclast precursors [128]. ANXA2 expression is elevated in all phases of MGUS, SMM, NDMM, and MM, as well as relapsed MM [88].

The LGALs1 gene exhibits Increased mRNA and protein expression that is uniform across inter-patient and inter-human myeloma cell lines (HMCLs) [129,130]. Subsequent analysis of primary CD138+ cells revealed that elevated LGALS1 expression is exclusively observed in NDMM patients as opposed to MGUS, SMM, MM relapsed MM, and healthy control groups [128,130]. According to Andersen et al., there was a marginally significant increase in LGALS1 protein in MM patients’ peripheral blood in comparison to healthy subjects. Nevertheless, the authors observed no significant association between the increased LGALS1 level and overall survival, treatment response, and clinical pathological parameters [131]. Moreover, the inhibition of LGALS1 in mouse models in vivo results in a reduction in tumor sizes, angiogenesis, and the emergence of bone lesions [130]. However, in 2017 Glavey et al. performed a proteomic characterization of the human multiple myeloma bone marrow extracellular matrix that revealed that MM patients who have elevated levels of ANXA2 and LGALS1 are associated with poor overall survival (OS) [128].

Another study revealed that alpha-enolase (ENO1) is significantly upregulated in the BM plasmacytoid dendritic cells (pDCs) of MM patients, which is linked with a reduced OS. Isocitrate Dehydrogenase (NAD(+)) 3 Catalytic Subunit Alpha (IDH3A) commences pathogenesis in MM via protein interaction with ENO1 [132]. According to proteome profiling, oxoglutarate dehydrogenase (OGDH) displayed a distinct expression profile in MM cells. OGDH is an enzyme component in the Krebs cycle that promotes cancer cell proliferation and survival [133]. In addition, ST3GAL6-AS1-mediated overexpression of long non-coding RNA (hnRNPA2B1) promoted MM cell adhesion and invasion [134]. In accordance with this, multipronged quantitative proteomic techniques on MM bone marrow interstitial fluid (BMIF) and serum identified five proteins, namely haptoglobin, kininogen 1, transferrin, apolipoprotein A1, and albumin, reflecting them as a useful diagnostic panel to improve understanding of MM pathogenesis [135].

Chanukuppa et al. employed a variety of bioinformatics tools, including Ingenuity Pathway Analysis (IPA), Protein Analysis through Evolutionary Relationships (PANTHER), Search Tool for the Retrieval of Interacting Genes/Proteins (STRING), and Database for Annotation, Visualization, and Integrated Discovery (DAVID), to investigate protein–protein interactions and their networking pathways. They proved that marginal zone B and B1 cell specific protein (MZB1) were expressed substantially differently in MM mononuclear cells (MNCs) isolated from MM patients. Upregulation of the MZB1 protein is one of the indicators for the pathogenesis and progression of MM and is anticipated to be a promising biomarker in the future [136].

Liquid chromatography-tandem mass spectrometry (LC-MS/MS) was used to discover metastasis associated with family member 2 (MTA2) and protein argonaute-2 (AGO2) upregulation. The Kyoto Encyclopedia of Genes and Genomes (KEGG) enrichment-based protein–protein interaction (PPI) network study found that both are essential regulators for multiple signaling pathways governing apoptosis, including miRNAs in cancer, ubiquitin-mediated proteolysis, the nucleosome remodeling and deacetylase (NuRD) complex, tumor protein P53 (p53), mitogen-activated protein kinases (MAPKs), and forkhead Box, type O (FOXO) signaling pathways. Upregulation of AGO2 is associated with a shorter survival time in MM patients; however, there was no significant difference in survival time between patients with high and low MTA2 expression levels. Patients with MGUS, NDMM, and relapsed and refractory MM (RRMM) had higher levels of MTA2 and AGO2, whereas VGPR and control patients had no significant fluctuations [137].

MM cell lines’ (MM1.S) responses to velcade, doxorubicin, and dexamethasone (VDD) treatment were investigated using 8-plex-iTRAQ-based mass spectrometry, which revealed 48 upregulated and 98 downregulated proteins. Proteins and enzymes such as macrophage inhibitory factor (MIF), stathmin (STMN1), Golgi resident protein (ACBD3), sec 23B, 7 kDa heat shock protein (HSPA8), importin-7 (IPO7), triosephosphate isomerase (TPI1), enoyl coa hydratase (ECHS1), and phosphoglycerate kinase (PGK1) were regulated in opposite directions [138].

Protein translation is observed to be more than 2.5-fold higher in primary MM cells than in healthy plasma cells [139]. Combining protein translation inhibitors, such as omacetaxine, with ImiD results in a twofold suppression of the Interferon regulatory factor 4/c-proto-oncogene (IRF4/c-MYC) pathway, resulting in lower MM survival. This was most noticeable in ex vivo treatments, as omacetaxine killed more than half of the MM cells. Protein translation inhibitors, as abovementioned, have the potential to be a novel drug in the treatment of MM [139]. To circumvent antigen evasion, clonal expansion, and/or disease progression in MM, the use of multiple targeted antigens could improve therapy outcomes while minimizing toxicity to normal cells. Kose et al. used a combination of surface proteomic and transcriptome approaches on six MM cell lines to identify proteins with significant differences in expression, including Fc receptor-like protein 5 (FCRL5), B-cell maturation antigeIL6Rn (BCMA), and intercellular adhesion molecule 2 (ICAM2) in all samples and interleukin 6 receptor (IL6R), B-type endothelin receptor (ETRB), solute carrier organic anion transporter family member 5A1 (SLCO5A1), and phosphatase of regenerating liver 3 (PRL3) in >60%, which were verified using flow cytometry. Furthermore, using a single cell RNA-sequencing assay, they discovered that clonal cells in bone marrow primarily expressed ETRB, which has potential prognostic significance for MM [140].

To gain better insight into the mechanisms that are responsible for the progression of multiple myeloma, a study was conducted on CD138-positive plasma that was recruited from the bone marrow of patients with MGUS, SMM, and MM. This study was driven by the mechanism by which MM cells adapt to the hypoxic environment in the bone marrow. Interestingly, proteomic analysis via LC-MS/MS assay discovered 6218 responsible proteins, implying that the biological pathways encounter multiple changes in an attempt to adapt to hypoxia pressure in MM cells along the path of disease progression, including regulation in Cytochrome c oxidase subunit 6C (COX6C) and Mitochondrially Encoded Cytochrome C Oxidase II (MT). The same study discovered a protein cluster implicated in apoptosis regulation, such as Bcl-2-associated X (BAX) and cysteine-aspartic acid protease 10 (CASP10) (regulated at the advanced MM stage), as well as proteins involved in regulating endoplasmic reticulum (ER) activities, such as Peroxiredoxin-4 (PRDX4) and thioredoxin domain-containing protein 11 (TXNDC11). Stromal cell-derived factor 2-like protein 1 (SDF2L1), myeloid-derived growth factor (MYDGF), stromal cell-derived factor 4 (SDF4), RCN1, and thioredoxin domain-containing protein 5 (TXNDC5) protein upregulation and downregulation in BAX, TNF receptor-associated factor 2 (TRAF2), inositol 1,4,5-trisphosphate receptor type 1 (ITPR1 ITPR1), and PRA1 family protein 2 (PRAF2) proteins were found to be involved in inhibiting the ER stress-induced apoptosis pathway. Furthermore, new protein candidates in the immune response evasion pathway, including stabilin-1 (STAB1), single Ig IL-1-related receptor (SIGIRR), dual adapter for phosphotyrosine and 3-phosphotyrosine and 3-phosphoinositide (DAPP1), and V-type immunoglobulin domain-containing suppressor of T-cell activation (VSIR), were identified as hijacking immunological responses in MM cells as well. In addition, the expression of SLAM7, CD46, SIGIRR, and SDC1 (CD138) demonstrated significant deregulation across all MM phases [141].

Xiao et al. utilized 2-DE and MALDI-TOF/TOF analysis, which resulted in the identification of 43 significantly distinct proteins. These proteins were further characterized into 16 groups due to their functional properties, such as cell adhesion molecules, chaperones, cytoskeletal proteins, defense immunity proteins, hydrolases, isomerases, kinases, ligases, nucleic acid binding, and oxidoreductases. By using western blot analysis, additional verification was received for six of them: profilin-1, glutathione peroxidase 1, annexin A1, proteasome subunit alpha type-5 (PSMA5), proteasome activator complex subunit 2 (PSME2), and proteasome subunit beta type-10 (PSMB10) [142]. Profilin-1 is believed to function as a negative regulator of MM aggressiveness and is classified as a tumor-suppressor protein [143]. It is also involved in a diverse range of cellular functions, including proliferation, migration, endocytosis, mRNA splicing, and gene transcription [144]. Upregulation of PSME2, PSMA5, and PSMB10 was observed in MM cell lines. This reflects the important roles that these proteins play in inhibiting short-lived regulatory proteins controlling the cell cycle mechanism, receptors on the cell surface, modulation of ion channels, and antigen attachment. Upregulation of proteosomal proteins was also noticed in breast cancer [145], leukemia [146], and renal carcinoma [147]. In MM, changes to proteosomal proteins promote proteasomal breakdown processes. The upregulation of protein in MM was also seen in cytochrome c oxidase, glutathione peroxidase 1 (GPX1), and four diverse types of dehydrogenases. These enzymes facilitated various cellular mechanisms and regulated reduction/oxidation (redox), which is essential for the maintenance of homeostasis [148]. An increase in the activity of antioxidant enzymes such as GPX1 was also recognized in another study conducted by Kuku et al. [149]. Annexin and S100 families, which are involved with calcium ion-binding proteins, were also strongly regulated in MM, as a prior study also reported downregulation of S100A9 and S100A12 genes in MM [10]. On the other hand, an increasing trend was observed in annexin A1 and annexin A11, which have an effect on several different cellular processes, including invasiveness, cell proliferation, motility, and signaling pathways [150].

Numerous studies conducted in the past have provided evidence of the existence of a sizable data pool for the identification of proteins in active MM. As a consequence of this, this review presents a condensed description of the examination of protein–protein interactions in active MM disease by utilizing STRING v11.5 (https://string-db.org/, accessed on 3 March 2023). Analysis of the 73 significantly regulated proteins (recruited from this review) is shown in Figure 2a; the cutoff score for the STRING database was adjusted to >0.4, and the proteins were grouped into three distinct clusters (Figure 2b–e). Table 2 summarizes the list of individual proteins involved in each cluster, and Table 3 illustrates the functional enrichments in protein networks according to the top three most significant biological processes represented in gene ontologies. Cluster 1 represents a group of proteins involved in multiple biological processes, including negative regulation of blood coagulation, negative regulation of multicellular organismal processes, fibrinolysis, and regulation of homotypic cell–cell adhesion. Cluster 2 consists of proteins involved in the ATP metabolic process, oxidation–reduction process, glycolytic process and generation of precursor metabolites and energy. Cluster 3 demonstrated proteins responsible for antigen processing and presentation of peptide antigen via MHC Class I, antigen processing and presentation of exogenous peptide antigen, response to endoplasmic reticulum stress, and cellular response to topologically incorrect proteins. MGUS precursors underwent extensive mutational events at a magnitude of 0.1 to 10 per megabase, along with genetic alterations that affected one or more genes, including aneuploidy, chromosomal translocations, single nucleotide variants, small insertions and deletions, and copy number variants in the state transition to SMM [94,95,96]. It was reported that the higher progression risk of SMM to MM was 10% per year, whereas the progression risk of MGUS was only 1% [97].

In Table 3, Cluster 1 represents proteins involved in multiple biological processes, including negative regulation of blood coagulation, negative regulation of multicellular organismal processes, fibrinolysis, and regulation of homotypic cell–cell adhesion. Cluster 2 consists of proteins involved in the ATP metabolic process, the oxidation–reduction process, the glycolytic process and the generation of precursor metabolites and energy. Cluster 3 demonstrates proteins responsible for antigen processing and presentation of peptide antigen via MHC Class I, antigen processing and presentation of exogenous peptide antigen, response to endoplasmic reticulum stress, and cellular response to topologically incorrect proteins.

### 4.4. Myeloma Bone Disease

Myeloma bone disease is a typical occurrence among myeloma patients and can be diagnosed in 70 percent of NDMM patients. Patients with MM who have bone disease suffer from multiple complications, the most significant of which are bone pain (80% of cases), bone fractures (60%), hypercalcemia (15%), and spinal cord compression (3%), all of which contribute significantly to the morbidity and mortality associated with the disease [151]. Previous research employing SELDI-TOF MS revealed a set of four unique biomarkers that could be used to monitor myeloma bone disease in the future [152]. Within a few years, proteome profiling using label-free mass spectrometry revealed a considerable increase in C4 and serum paraoxonase/arylesterase 1 concentrations from MGUS/SMM to severe bone disease [153]. Significantly different serum proteome profiles were studied in MM patients with various degrees of bone disease, describing the interplay between binding and inhibition. Multiple proteins associated with MM bone disease have been identified using a label-free mass spectrometry technique, including enzymes, extracellular matrix glycoproteins, and complement system components. The central protein network displays the most downregulated expression in alpha-1-antitrypsin (SERPINA1), albumin (ALB), α-2-macroglobulin (A2M), coagulation factor V (F5), fibronectin (FN1), plasminogen (PLG), kininogen 1 (KNG1), and platelet factor 4 (PF4), with only fibronectin showing downregulation (FN1) [153].

### 4.5. Relapse/Refractory MM

A distinct extracellular matrix (ECM) profile was discovered in relapsed MM patients, with 25 significantly elevated proteins including 10 matrisome core proteins and 15 matrisome associate proteins. MGUS and MM patients exhibit a reduction in collagen and other fibrillar ECM glycoproteins, such as fibronectin, when compared to healthy individuals. The presence of specific collagens and matrix metalloproteinases (MMPs) (COL1A1, COL1A2, COL3A1, COL5A1, MMP8 and MMP9) in both healthy donors and MM patients [88] indicates early ECM remodeling leading to MM progression [154]. Alterations in ECM contributed to several stages of metastasis. Furthermore, interactions with the bone marrow microenvironment led to the progression of MM and enhanced a tumorigenic microenvironment [155].

Proteinase inhibitor 9 (SERPINB9) and 34 additional proteins with differential expression in patients with bortezomib-resistant RRMM compared to NDMM patients were discovered using tandem mass tag-mass spectrometry (TMT-MS). Overall, the discovered proteins demonstrated roles in regulating cellular metabolism, apoptosis, programmed cell death, lymphocyte-mediated immunity, and defensive response pathways in RRMM [156]. SERPINB9 is the most established serine protease inhibitor (serpin), and inhibits the serine protease granzyme B (GrB) [157] in the granules of cytotoxic T lymphocytes (CTLs) via the perforin pathway [158,159], antigen-presenting cells, and natural killing cells [160,161], thereby manipulating the apoptotic cascade [162]. Additionally, SERPINB9 induces resistance to CTL-mediated cell death and cancer cell immune response escape [163].

Numerous upregulated protein trends were observed in refractory MM patients, including proteasome activator complex subunit 1 (PSME1), PSME2, heat shock protein 90 (HSP90), HSPA9, stress-induced-phosphoprotein 1, nucleophosmin, and protein disulfide-isomerase, which significantly differentiated them from patients with very good partial or complete responses. Furthermore, differences in expression were found in proteins involved in oxidative stress and redox homeostasis (thioredoxin (TXN)). Protein downregulation in apoptosis and inflammation-related proteins was seen in bortezomib therapy failure patients. Likewise, MM patients who were resistant to bortezomib had changes in their proteome profile affecting apolipoprotein C1, complement components, and sulfhydryl oxidase 1, all of which play key roles in modulating hydrolase activity and cellular reactions to stimuli [164].

### 4.6. Extramedullary MM

The aggressive stage of MM is characterized by extramedullary MM (EMM), which was reported in roughly 7% of NDMM and increased to 30% of patients with relapse [165]. After a series of molecular alterations, the clonal cells of the MM disease were able to escape from the bone marrow and migrate via the circulatory system to access and grow independently in other organs and tissues [166]. Zatula et al. employed super-SILAC quantitative proteome profiling to study alterations in proteome architecture in MM and secondary plasma cell leukemia (sPCL) patients and discovered that a few potential protein markers could be targeted by currently offered small molecule therapies. Interestingly, proteome change was distinguished by the vast number of proteins affected (*n* = 795). Patients with plasma cell leukemia (PCL) had metabolic changes in aerobic glycolysis, and downregulation of enzymes responsible for glycan production led to changes in surface receptor glycosylation pathways. They also discovered that no substantial change in genomic 5-methylcytosine or 5-hydroxymethylcytosine was observed at either stage, indicating that epigenetic dysregulation is not the primary driver in the progression from MM to PCL [167].

A label-free liquid chromatography mass spectrometric approach identified 21 proteins with significantly different expression in EMM patients compared to MM patients. Antibody-based ELISAs were used to validate six proteins (vascular cell adhesion molecule 1 (VCAM1), hepatocyte growth factor activator (HGFA), pigment epithelial-derived factor (PEDF), α-2-macroglobulin (A2M), cholinesterase (BCHE), and aminopeptidase N (CD13)). Consequently, ROC studies indicate that VCAM1, HGFA, and PEDF have excellent discriminatory power for predicting EMM [168]. Another quantitative MS-based proteomic method revealed that 275 and 271 proteins in EMM were significantly upregulated and downregulated, respectively. Rho-associated protein kinase 2 (ROCK2), Ras-related C3 botulinum toxin substrate 1 (Rac1), and platelet endothelial cell adhesion molecule (PECAM-1) were among the proteins that were upregulated. Interestingly, overexpression of proteins associated with leukocyte transendothelial invasion was also discovered [169].

## 5. Investigating Biomarkers in Drug-Resistant MM

The most challenging part of treating MM patients is overcoming treatment resistance, which is most likely caused by a complex interaction between clonal heterogeneity and therapeutic pressure. The inevitable emergence of multi-drug resistance in MM contributes to the battle in drug research to identify the protein phenotype responsible for clone resistance [170]. Proteomics technology makes it possible to find out which proteins directly affect the biological processes that lead to the formation of resistant sub-clones.

Quantitative proteomics demonstrate that myristoylated alanine-rich C-kinase substrate (MARCKS) overexpression and phosphorylation are consistent with bortezomib-resistant MM cell lines [171]. LC-MS/MS analysis has also revealed that clusterin (CLU), angiogenin (ANG), and complement 1Q (C1Q) biomarkers are associated with the bortezomib response [172]. Another study found higher expression of apolipoprotein C-I and apolipoprotein C-I’ in bortezomib-resistant MM patients using mass spectrometry [173]. iTRAQ and label-free quantitative mass spectrometry detected 118 proteins (35 up-regulated and 83 down-regulated) that have significantly different expression in a patient with relapsed refractory MM who had a relatively low response to bortezomib, doxorubicin, and dexamethasone (PAD) chemotherapy and achieved a very good partial response (VGPR). The proteins that were discovered are those that are responsible for proteasome activity, responsiveness to oxidative stress, the defensive response, and the regulation of apoptosis [174]. The Frassanito group reported that the oxidative stress response and pro-survival autophagy pathways were activated in bortezomib-resistant MM patients, based on a proteomic study of bone marrow stromal cells (BMSCs) [175]. Metabolomic analysis of ANBL-6 cell lines resistant to bortezomib revealed a significant change in purines and pyrimidines anabolism and catabolism, as well as numerous forms of coenzyme A (CoAs) [176]. In addition, the same analysis demonstrated that hypoxia-inducible factor 1 and its target gene lactate dehydrogenase A are responsible for hypoxia-induced bortezomib resistance in MM [177]. Integrated quantitative tandem mass tag (TMT)-based proteome and phosphoproteomic profiling revealed that CDK6 protein is upregulated in in vitro-generated lenalidomide-resistant MM.1S and LP-1 cells, suggesting that it is a potential target for IMiD-resistant MM.1S [178].

Proteomics analysis and bioinformatics tools proved that the exportin-1 (XPO1) protein and its network contributed to emerging bortezomib resistance in the RPMI 8226 human MM cell line. The cluster of interaction proteins includes structural maintenance of chromosomes protein 1A (SMC1A), regulator of chromosome condensation 2 (RCC2), CSE1, nuclear pore complex protein Nup88 (NUP88), nuclear pore complex protein Nup50 (NUP50), nucleoprotein TPR (TPR), heat shock protein 70 kDa 14 (HSPA14), dynein light chain 1 (DYNLL1), double-strand-break repair protein rad21 homolog (RAD21), and E3 SUMO-protein ligase RanBP2 (RANBP2) [179]. XPO1 is a member of the exportin protein family, also known as the (chromosome region maintenance 1) protein, which has been linked to a variety of cancers, including hematological malignancies [180,181,182,183]. XPO1 is recognized as a nuclear transporter of various RNAs and protein loads from the nucleus to the cytoplasm [184,185], and disturbance in this transport channel is typically associated with cancer emergence [186], where extreme nuclear export promotes cancer progression and treatment resistance [187].

Karen et al. demonstrated that treatment with dexamethasone altered the proteome profile of the MM.1S cell line, resulting in the downregulation of ten proteins involved in cell proliferation or survival pathways, including proliferating cell nuclear antigen (PCNA) (D15), eukaryotic translation elongation factor 1 gamma (D13), inosine-5′-monophosphate dehydrogenase 2 (D7), and SUMO-1 activating enzyme subunit 1 (D14). In contrast, three proteins were upregulated, including Peptidyl-prolyl cis-trans isomerase (FKBP5) (U2 and U3), which is implicated in protein folding and trafficking [188], and prolyl 4-hydroxylase alpha-1 subunit precursor (4-PH alpha-1) (U1), an essential component of post-translational protein modification [189]. A mass spectrometry-based quantitative global proteomic, PPIs and gene ontology (GO) analysis uncovered that both upregulated proteins identified in AMO-BTZ (AMO-1 myeloma cell line resistant to bortezomib) and AMO-CFZ (AMO-1 myeloma cell line resistant to carfilzomib) are associated with protein catabolism, redox control, and protein folding. In contrast, the roles of downregulated protein clusters include transcription, translation, differentiation, apoptosis, and structural and cytoskeletal functions [190].

Additionally, proteomics analysis of NDMM patients who did not respond to thalidomide treatment revealed a significant increase in serum amyloid-A protein (SAA), 2M, vitamin D-binding protein (VDB), and zinc-α-2-glycoprotein (ZAG) concentrations [191]. Furthermore, using a nano LC-ESI-MS/MS method, Zhou et al. discovered that Runt-related transcription factors 1 and 3 (RUNX1 and RUNX3) are associated with lenalidomide resistance in MM. Following binding inhibition of IKZF1 and IKZF3, both transcription factors network with cereblon, resulting in additional suppression of cereblon-dependent IKZF ubiquitination and ultimately escape from proteasome degradation [192]. Proteomic profiling using 2D-PAGE and MALDI-TOF/TOF distinguished the dexamethasone-sensitive MM.1R cell line from the dexamethasone-resistant MM.1R cell line by revealing a substantial overexpression of FKBP5, which is responsible for the apoptotic pathway [189]. Further proteomics analysis revealed that exposing the MM1.R cell line to elevated circulating extracellular vehicles (EVs), which is accompanied by an increase in CD44 expression [193], induces IL-6 production and cell adhesion-mediated drug resistance (CAM-DR) [194]. Furthermore, CD44 promotes dexamethasone resistance in MM by attaching to extracellular matrix proteins such as hyaluronan (HA) [195]. Unfortunately, patients with poor overall survival have serum CD44 concentrations greater than 280 ng/mL [196].

## 6. Therapeutic Agents to Target Protein Signatures

Numerous therapeutic agents have been studied that target protein signatures identified through proteomic analysis in the context of multiple myeloma [197]. The following therapeutic agents are currently available. Bortezomib (Velcade; Takeda Pharmaceuticals, Cambridge, MA, USA) was granted approval by the U.S. Food and Drug Administration (FDA) in 2003, marking the first proteasome inhibitor authorized for use as a third-line treatment of multiple myeloma (MM) [198]. The advent of bortezomib has brought about an important advancement in the management of MM. Presently, bortezomib is sanctioned as a primary therapeutic option for MM and mantle cell lymphoma [199]. Bortezomib is a pharmacological agent classified as a proteasome inhibitor that selectively targets the 26S proteasome complex. The aforementioned phenomenon hinders the processes of protein degradation, transcription factor activation, and cell-cycle regulation and triggers programmed cell death in cells affected by multiple myeloma [200,201]. Bortezomib has obtained regulatory approval for the therapeutic management of multiple myeloma and is frequently administered in conjunction with other drugs. The utilization and efficacy of bortezomib in the management of multiple myeloma were investigated by Loke et al. The study involved patients who were administered with standard regimens such as Bortezomib, Melphalan and Prednisolone (VMP), Bortezomib, Cyclophosphamide, and Dexamethasone (CVD), and Bortezomib and Dexamethasone VD, and received Bortezomib either once weekly or twice weekly. It was reported that the overall response rate was 81%. Approximately 53% of the patients were unable to complete their intended treatment regimen due to various reasons, such as toxicity (30%), suboptimal response or disease progression (15%), or mortality during treatment (8%). The study reported a median overall survival of 40.7 months and a median progression-free survival of 17.7 months [202].

Another proteasome inhibitor (PI), Carfilzomib (formerly PR-171), belongs to the epoxyketone class of proteasome inhibitors and exhibits structural and functional variations from BTZ [203]. CFZ has been observed to exhibit strong potential for binding to the i20S proteasome through its highly specific inhibition of CT-L activity. As a consequence, CFZ induces antiproliferative and proapoptotic outcomes in multiple myeloma cell lines [204]. It is noteworthy that CFZ demonstrates an enhanced and prolonged response by means of irreversible proteasome inhibition, in comparison to BTZ, which functions as a reversible proteasome inhibitor [203]. Furthermore, the pharmacophore of epoxyketone exhibits a remarkable capacity for precise targeting of the NH2-terminal threonine residue, thereby enhancing specificity towards the proteasome and ultimately resulting in the irreversible inhibition of enzyme activity [204,205]. The utilization of carfilzomib was observed in combination with lenalidomide and dexamethasone, as well as independently, in patients with relapsed and/or refractory multiple myeloma and diverse levels of renal capacity [205,206]. In July 2012, the FDA granted approval for the use of Carfilzomib (Kyprolis^®^) as a monotherapy for the management of MM in patients with refractory disease [207]. This approval was reserved for patients who had undergone at least two prior lines of therapy and had exhibited disease progression within 60 days of completing their most recent therapy [208]. The ASPIRE study completed an investigation into the comparative overall survival rate of carfilzomib, lenalidomide, and dexamethasone (KRd) versus lenalidomide plus dexamethasone (Rd) for patients diagnosed with relapsed or refractory multiple myeloma. Interestingly, KR exhibited a statistically significant and clinically relevant decrease in mortality risk by enhancing survival for a duration of 7.9 months [209]. In 2016, the FDA granted approval for the use of Carfilzomib in combination with dexamethasone. This treatment is intended to treat patients with relapsed and/or refractory multiple myeloma who have undergone one to three prior therapies [210]. The decision was established on the basis of information obtained from the phase 3 ENDEAVOR clinical trial, which exhibited a progression-free survival benefit for carfilzomib in combination with dexamethasone in contrast to bortezomib in combination with dexamethasone [211]. As a result, the approval status of carfilzomib’s monotherapy was upgraded from accelerated approval to full FDA approval in this particular context [211].

Ixazomib is a second-generation PI and an orally administered PI that specifically targets the 20S proteasome [212]. Minarik et al. provide evidence that the addition of ixazomib to the lenalidomide–dexamethasone (RD) treatment regimen results in significant enhancements in critical survival outcomes among patients diagnosed with relapsed and refractory multiple myeloma (RRMM) in a standard clinical environment. Patients who received IRD treatment exhibited enhanced PFS and OS outcomes, with a median duration of 36.6 months and 26.0 months, respectively. Nonetheless, the administration of IRD treatment did not confer any advantages to patients exhibiting extramedullary disease, as evidenced by a median PFS of 6.5 months [213].

Additionally, daratumumab is a monoclonal antibody that specifically targets CD38, a protein that is prominently expressed on the surface of multiple myeloma cells [214,215]. It facilitates diverse immune-mediated mechanisms to elicit apoptosis of myeloma cells [216]. Daratumumab has received regulatory approval for the management of multiple myeloma, both as a monotherapy and in conjunction with other therapeutic modalities. Clinical trial data evidenced that the administration of daratumumab in patients with multiple myeloma resulted in enhanced survival rates among specific demographic groups. In the CASTOR study, the efficacy and safety of daratumumab in combination with bortezomib and dexamethasone were evaluated in comparison to bortezomib and dexamethasone alone, for patients suffering from relapsed MM. The findings indicate that the incorporation of daratumumab resulted in a noteworthy enhancement in the overall survival rate. The daratumumab group exhibited an indeterminate median overall survival, while the control group demonstrated a median overall survival of 40.0 months [217]. Additionally, the POLLUX study was conducted to assess the effectiveness and safety of daratumumab when used in combination with lenalidomide and dexamethasone, as compared to the use of lenalidomide and dexamethasone alone, in patients suffering from relapsed or refractory multiple myeloma. This clinical trial indicates a noteworthy enhancement in the survival rate as a result of daratumumab administration. The daratumumab group exhibited an indeterminate median overall survival, whereas the control group demonstrated a median overall survival of 44.3 months [218]. Another clinical trial, the MAIA study, conducted a clinical evaluation of the effectiveness and safety of daratumumab in conjunction with lenalidomide and dexamethasone, as compared to the administration of lenalidomide and dexamethasone alone, among individuals diagnosed with multiple myeloma who were deemed ineligible for transplant. The incorporation of daratumumab led to a notable enhancement in overall survival. The daratumumab group demonstrated an indeterminate median overall survival, while the control group exhibited a median overall survival of 68.3 months [219].

Elotuzumab is a monoclonal antibody that specifically targets SLAMF7 (CS1), a cell surface protein that is expressed on myeloma cells [220]. The drug has been sanctioned for the management of multiple myeloma in conjunction with other agents, as it amplifies immune-mediated cytotoxicity against myeloma cells [221].

Venetoclax is a diminutive molecule inhibitor that specifically targets BCL-2, an anti-apoptotic protein that is frequently overexpressed in multiple myeloma cells [222]. The medication has been authorized for the management of relapsed or refractory multiple myeloma in conjunction with other drugs by inducing apoptosis [223]. It is noteworthy that the domain of multiple myeloma is subject to persistent research and development, with the emergence of novel therapeutic agents that target specific protein signatures.

## 7. Conclusions

The high heterogeneity of MM, combined with the fact that it is typically diagnosed in advanced stages, makes it extremely difficult to manage. MM is also linked to increased proteome complexity and numerous post-translational modification (PTM) events, which change the structure, function, and/or location of proteins. PTM events are crucial in MM as a degradative strategy in protein degradation or turnover that helps cells maintain their physiological homeostasis. Proteomic technology has advanced dramatically, allowing for the identification of prospective protein panels that could aid in the future management of MM disease and hence boost MM patient survival rates. To date, proteomics techniques have been used in conjunction with more sensitive and powerful mass spectrophotometers, extensive protein databases, and bioinformatic software to conduct significant research on proteome changes in a variety of diseases. There are currently no proteomics approaches that have discovered the complete human proteome. In the context of MM, investigating proteomic changes in MM PCs and comparing them to normal PCs is an essential goal for elucidating the most likely protein biomarkers involved in MM etiology. Along with this discovery, major molecular mechanisms and pathways implicated in MM progression, as well as possible biomarkers for therapeutic use, were also discovered. Currently, the application of proteomics in MM has begun; however, it still requires extensive validation and follow-up biological and clinical research. Future translational research is also in demand to translate these troubling foundational findings into novel prognostic, diagnostic, and therapeutic tools that can enhance MM treatment and outcomes.

## Figures and Tables

**Figure 1 diagnostics-13-02328-f001:**
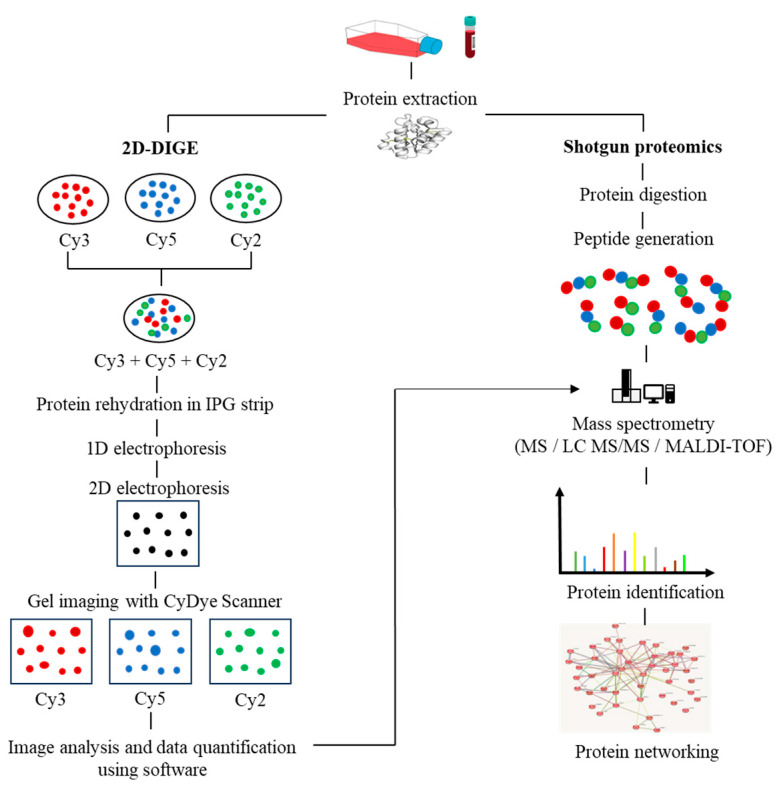
The workflow in proteomics experiments incorporates protein 2D-DIGE electrophoresis and shotgun mass spectrometry.

**Figure 2 diagnostics-13-02328-f002:**
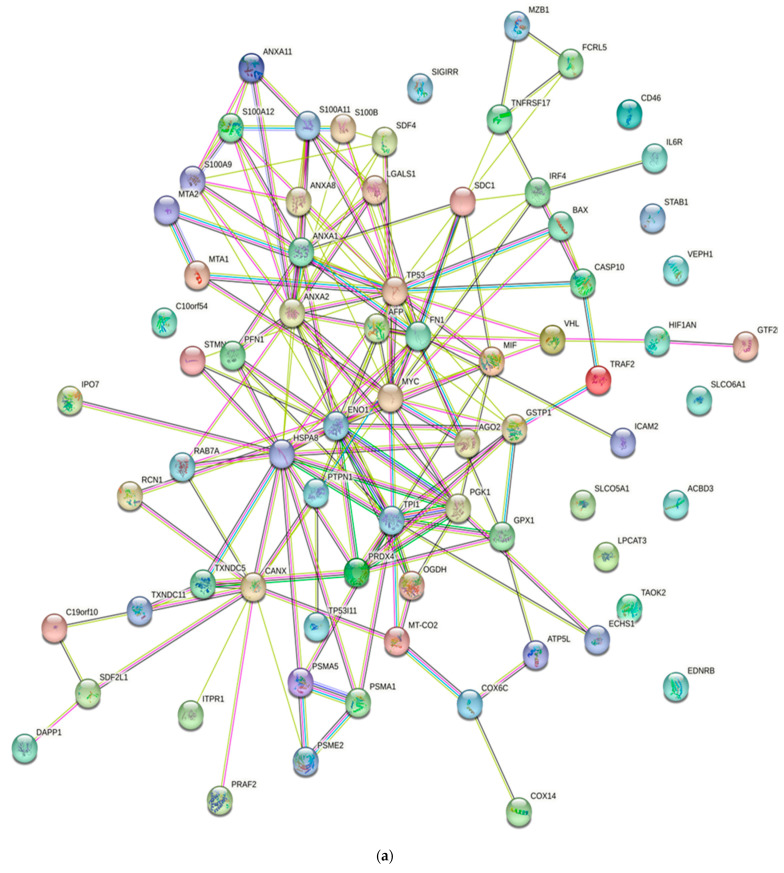
Protein–protein interaction network in active MM disease. Nodes represent proteins and edges represent protein–protein associations. The edges are composed of known and predicted interactions, among other forms of interaction, including co-expression (STRING https://string-db.org/) accessed on 3 March 2023. (**a**) Full STRING network. The edges indicate both functional and physical protein associations. (**b**) Full STRING network in cluster. Protein–protein networking in Cluster 1 (**c**), Cluster 2 (**d**), and Cluster 3 (**e**). The full STRING analysis report can be found at https://version-11-5.string-db.org/cgi/network?networkId=bwimO6bRnH24 accessed on 3 March 2023.

**Table 1 diagnostics-13-02328-t001:** Proteomic technologies for protein profiling.

Technique	Method Description
2-dimensional polyacrylamide gel electrophoresis (2D-PAGE)	The protein content of a sample is resolved on a gel in two dimensions according to mass and charge; the gels are stained, and the spot intensities in the samples are analyzed among the multiple gels.
2D-DIGE	Each protein sample of interest is labelled with a different fluorophore (Cy3, Cy5, or Cy2) that binds covalently to the epsilon amino group of lysine residues.
Protein microarrays	Direct labelling or labelled secondary antibodies are used to identify bound proteins once targeted proteins in one sample bind to probes on a “forward” microarray, and vice versa for “reverse” microarrays.
Surface-enhanced laser desorption/ ionization time-of-flight mass spectrometry (SELDI-TOF MS)	Retention chromatography and mass spectrometry principles are combined, offering a fast, high-throughput, and relatively sensitive screening approach for complicated protein samples. Proteins can also be separated, detected, and analyzed at the femtomole level straight from biological materials. This allows for the discovery of many analytes and the analysis of many diverse samples while studying multiple biological variables at the same time.
Matrix-assisted laser desorption/ ionization-time of flight mass spectrometry (MALDI-TOF MS)	Application of a protein mixture onto a gold plate, desorption of proteins from the plate using laser energy, and determination of the protein masses, with comparison of peak intensities between several different samples.
Liquid Chromatography with tandem mass spectrometry (LC-MS-MS)	Separation of a mixture of peptides (derived from trypsin-catalyzed protein digestion) through one-, two-, or three-dimensional LC and determination of peptide masses through MS-MS.
Isotope-coded affinity tag (ICAT)	Chemical tagging of proteins on cysteine residues with a heavy or light stable isotope; after labelling samples are combined, proteins are digested with trypsin, and tagged peptides are extracted via affinity chromatography; both samples are then concomitantly analyzed using LC-MS-MS.
Multiplexed Isobaric Tagging Technology for Relative Quantitation (iTRAQ)	A shotgun-based quantitation technique. Employs a multiplexed isobaric chemical tagging reagent that enables the multiplexing of two to eight protein samples and generates identical MS-MS sequencing ions for all eight variants of the same derivatized tryptic peptide.
SILAC (Stable Isotope Labeling by Amino Acids in Cell Culture)	Encompasses the practice of cultivating cells in a medium that comprises isotopically labelled amino acids. The isotopes utilized are generally heavy variants of essential amino acids, such as ^13^C- or ^15^N-labelled versions. The process of labelling induces a mass shift in the proteome, thereby facilitating precise quantification and comparison of protein expression levels across diverse conditions or samples.
Immunohistochemistry (IHC)	A diagnostic technique for visualizing and detecting particular proteins within tissue specimens. This technique facilitates the analysis of protein localization, interaction, and expression patterns with a high degree of spatial resolution.
Flow cytometry	The process of scrutinizing and measuring diverse physical and chemical attributes of individual cells or particles in a fluid suspension; is commonly referred to as a cellular or particle analysis technique. The technique facilitates the concurrent assessment of various parameters such as cell surface markers, intracellular proteins, DNA content, and cell viability.
Aptamer-based proteomics platforms	A novel technological approach involving the utilization of aptamers, which are short, single-stranded nucleic acids, as affinity reagents for identifying and analyzing proteins. The process of systematic evolution of ligands by exponential enrichment (SELEX) is utilized to generate aptamers that exhibit high affinity and selectivity towards target proteins. These aptamers are designed to specifically bind to the target proteins.
Singleplex ELISA (enzyme-linked immunosorbent assay (ELISA)	Also referred to as standard or traditional ELISA; a common methodology employed for the identification and measurement of an exclusive protein within a specimen.
Multiplex ELISA	A novel revision of the conventional enzyme-linked immunosorbent assay (ELISA) that enables the concurrent identification and measurement of numerous proteins within a solitary specimen. The utilization of multiplex ELISA allows for the investigation of multiple analytes present in intricate biological specimens, thereby facilitating a more exhaustive evaluation in contrast to singleplex ELISA.

**Table 2 diagnostics-13-02328-t002:** A total of 73 proteins retrieved from this review were further categorized into three clusters using kmeans clustering in STRING software. Cluster 1 consists of 19 proteins, Cluster 2 consists of 37 proteins and Cluster 3 consists of 17 proteins.

	Cluster Id	Gene Count	Protein Names
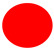	Cluster 1	19	ACBD3, ATP5L, COX14, COX6C, ECHS1, EDNRB, ENO1, GTF2F2, HIF1AN, ICAM2, LPCAT3, OGDH, PGK1, PRDX4, SLCO5A1, SLCO6A1, TAOK2, TPI1, TRAF2
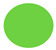	Cluster 2	37	AFP, AGO2, ANXA1, ANXA11, ANXA2, ANXA8, BAX, C10orf54, CASP10, CD46, FCRL5, FN1, GPX1, GSTP1, HSPA8, IL6R, IRF4, LGALS1, MIF, MTA1, MTA2, MYC, MZB1, PFN1, S100A11, S100A12, S100A9, S100B, SDC1, SDF4, SIGIRR, STAB1, STMN1, TNFRSF17, TP53, VEPH1, VHL
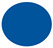	Cluster 3	17	C19orf10, CANX, DAPP1, IPO7, ITPR1, MT-CO2, PRAF2, PSMA1, PSMA5, PSME2, PTPN1, RAB7A, RCN1, SDF2L1, TP53I11, TXNDC11, TXNDC5

**Table 3 diagnostics-13-02328-t003:** Functional enrichments in the protein networks according to the top four most significant biological processes represented in gene ontologies with a significant false discovery rate.

Cluster	GO-Term	Biological Process	False Discovery Rate
1	GO:0030195	Negative regulation of blood coagulation	0.0020
	GO:0051241	Negative regulation of multicellular organismal process	0.0020
	GO:0042730	Fibrinolysis	0.0129
	GO:0034110	Regulation of homotypic cell–cell adhesion	0.0239
2	GO:0046034	ATP metabolic process	0.00044
	GO:0006096	Oxidation–reduction process	0.00054
	GO:0055114	Glycolytic process	0.0015
	GO:0006091	Generation of precursor metabolites and energy	0.0018
3	GO:0002474	Antigen processing and presentation of peptide antigen via MHC Class I	0.0044
	GO:0002478	Antigen processing and presentation of exogenous peptide antigen	0.0044
	GO:0034976	Response to endoplasmic reticulum stress	0.0047
	GO:0035967	Cellular response to topologically incorrect protein	0.0141

## Data Availability

Data are contained within the article.

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
