# Peer review of "Proteomic Alteration in the Progression of Multiple Myeloma: A Comprehensive Review"

_diagnostics, 2023, doi:10.3390/diagnostics13142328_

Round 1

Reviewer 1 Report

Overall, the quality of writing is average, but if the authors address the comments provided, this will be improved. Originality is good, there are very few reviews in the literature on this subject. Contribution is also good, as reviews like this are important in summarising research findings, in this case from a proteomics/MM perspective. It would be useful to have a section on available therapeutic agents to target protein signatures identified. This is becoming a standard approach in Ocology therapy. Why not in Myeloma? 

Under the section ‘Proteomics breakthrough in protein identification’ commentary on the following are worth including:

  • LFQ-DIA (data-independent acquisition) and LFQ-DDA (data-dependent acquisition) as frequently used quantification approaches using mass spectrometry

  • Cell surface capturing (CSC) – important approach to enrich for cell surface proteins – possible MM related therapeutic targets

  • Single cell proteomics and the use of Machine Learning (ML) and Deep Learning (DL) algorithms to identify protein signatures of disease.

Table 1 – needs to be expanded to include:

  • SILAC
  • Immunohistochemistry (IHC) & Immunofluorescence (IF)
  • Flow cytometry (FC)
  • Aptamer-based proteomics platforms
  • ELISA (single and multiplex approaches)

Under the section ‘Post-translational modifications in MM’ a section on enrichment approaches should be included, for example - the use of an antibody against the modifications (phosphorylation, acetylation, methylation, ubiquitination), in protease-digested lysates, as an enrichment strategy.

Figure 2/Tables 2&3 need more detail, what were the criteria for selecting these 73 proteins from the literature, were these proteins all from the same sample type (biofluid/cells), were these proteins all increased/decreased in abundance in their respect comparisons?  What were the comparison, e.g. MGUS v MM, etc …

Figure 1 needs to be changed, with a separate workflow for shotgun MS (including protein digestion, labelling) and for the 2D component, DIGE would be a more suitable example (more quantitative).

A lot of sentences are over long. Grammatical errors. 

Careful re reading, simplification of sentences and check with bibliography is required. 

Below are some examples of suggested corrections. 

Suggested Edits-

Introduction 

Line 22 in MM

Line 23 in MM

Line 40 MM evolves from

Line 42 (SMM), to symptomatic MM

Line 42,3 symptomatic MM (intramedullary and extramedullary Myeloma).

Line 45,6 delete autologous stem cell transplant-this is not a “novel” drug treatment. 

Line 44-median survival rate

2 Proteomics breakthrough in protein identification 

L91- Remarkably, significant changes in gene expression are not necessarily reflected at the protein level. 

Line 480

Relapsed/refractory MM

Line 514 Extramedullary MM

Reviewer 2 Report

This is a review article aimed to comprehensively summarize the findings of proteomic evaluation of plasma cell dyscrasias. The work is moderately comprehensive but notably does not source some of the most provocative findings (i.e “Patients with MM who have greater ANXA2 and LGALS1 levels are associated with poor overall survival (OS)” lines 287-288). Additionally, the author’s interpretation of the results summarized seems overly generalized. For example, they point to changes in the ECM of bone marrow in MM patients, but then states that these findings were verified in MM cell lines making this finding quite confusing. They also appropriately mention the discordance between genomic and proteomic findings but then extensively discuss miRNA findings (lines 180-195) without suggesting any proteomic correlation. Notably, the authors often generalize findings of proteomic analysis of patients with active MM without taking into context the role of treatment on their proteome which is somewhat misleading especially when comparing these finding to MGUS and SMM who are likely treatment naïve. Finally, some of the background information around myeloma is incorrect and contrary to the sources that are referenced. For example, the authors notes that MM accounts for 1% of all neoplastic disease and “> 10% all hematologic malignancies”, but the paper referenced would suggest that it is significantly higher. Similarly, the authors state that “the majority of patients are diagnosed late in the course of the disease” which may be the case in some areas but is not true in most populations. 

Moderate editing is required. 

Round 2

Reviewer 2 Report

Revisions are appropriate.